

# Winter thermodynamic vertical structure in the Arctic atmosphere linked to large scale circulation

Tiina Nygård[1], Michael Tjernstöm[2], Tuomas Naakka[1]

[1]Finnish Meteorological Institute, Helsinki, Finland

[2] Department of Meteorology, and Bolin Centre for Climate Research, Stockholm University, Stockholm, Sweden

*Correspondence to*: Tiina Nygård (tiina.nygard@fmi.fi)

**Abstract.** Thermodynamic profiles are affected by both the large scale dynamics and the local processes, such as radiation, cloud formation and turbulence. Based on ERA5 reanalysis, radiosoundings and cloud cover observations from winters 2009–2018, this study demonstrates manifold impacts of large scale circulation on temperature and specific humidity profiles in the

circumpolar Arctic north of 65°N. Characteristic wintertime circulation types are allocated using Self-Organizing Maps (SOMs). The study shows that influence of different large scale flows must be viewed as a progressing set of processes: (1) horizontal advection of heat and moisture, driven by circulation, lead to so-called first order effects on thermodynamic profiles and turbulent surface fluxes, and (2) the advection is followed by transformation of the air through various physical processes, causing second order effects. An example of second order effects is the associated cloud formation, which shifts the strongest

radiative cooling from the surface to the cloud top. The temperature and specific humidity profiles are most sensitive to large scale circulation over the Eurasian land west of 90°E and the Arctic Ocean sea ice, whereas impacts over North America and Greenland are more ambiguous. Eurasian land, between 90°E and 140° E, occasionally receives warm and moist air from the northern North Atlantic, which, with the support of radiative impacts of clouds, weaken the otherwise strong temperature and specific humidity inversions. Altitudes of maximum temperature and specific humidity in a profile, and their variability

between the circulation types, are good indicators of the depth of the layer impacted by surface–atmosphere processes interacting with the large scale circulation. Different circulation types typically cause variations of a few hundred meters to this altitude, and the layer impacted is deepest over north-eastern Eurasia and North America.

## 1 Introduction

Spatial distributions of atmospheric temperature and humidity in the wintertime Arctic are to a large extent controlled by the

predominantly negative surface net radiation budget and poleward energy transport by the atmosphere; in an average annual sense, this transport is necessary to balance the radiation loss to the space at the top of the atmosphere, for a stable climate. Horizontal advection of heat and moisture, dictated by large scale atmospheric circulation (Nygård et al., 2019; Papritz, 2020; Messori et al., 2018), have direct implications on distributions of temperature and humidity, but also affect other meteorological conditions, such as cloud formation and properties, and surface fluxes, in turn modifying temperature and humidity

distributions in the atmosphere. The thermodynamic characterization of the lower Arctic atmosphere hence requires understanding of both the local processes, such as radiation, cloud formation and turbulence, and the larger scale dynamics, responsible for the advection (Morrison et al., 2012).

The warmest anomalies in the wintertime Arctic are connected to horizontal transport of air from regions with higher
climatological potential temperature and to diabatic heating (Papritz, 2020). Especially, a meridionally-oriented circulation pattern, or cyclone track, favours intrusions of warm and moist air from the mid-latitudes into the Arctic (Messori et al., 2018; Woods et al., 2013; Fearon et al., 2021; Pithan et al., 2018). Such intrusions are associated with increased cloudiness, cloud water content and downward longwave radiation (Nygård et al., 2019; Liu et al., 2018; Park et al., 2015). The coldest anomalies in the central Arctic, on the other hand, occur after a prolonged period of uninterrupted radiative cooling, when the large scale
circulation pattern shelters the high Arctic from meridional air mass exchange (Papritz, 2020; Messori et al., 2018; Pithan et al., 2018) and cloud free conditions often prevail.

Due to the radiative cooling at the surface in clear sky conditions, surface-based temperature inversions are very common in winter over the snow or sea ice covered surfaces (Curry, 1983; Serreze et al., 1992; Devasthale et al., 2010). However, when
warm and moist air is advected from the lower latitudes into the Arctic, it experiences radiative cooling, leading to cloud formation (Pithan et al., 2014). In cloudy conditions, the strongest radiative cooling rapidly shifts from the surface to the cloud top, causing buoyancy production and mixing from turbulent overturning, while increased longwave radiation from the cloud base warms the surface (Tjernström and Graversen, 2009; Lemone et al., 2019). As a consequence, an elevated temperature inversion forms over a shallow but relatively well-mixed boundary layer. However, continued cooling of the cloud and related
phase changes from liquid to ice gradually reduce the emissivity of the cloud (Pithan et al., 2014). Finally, after cloud ice formation has caused the cloud to precipitate out, the clear state with a surface-based temperature inversion again prevails. These shifts between surface-based and elevated temperature inversions are rapid in winter (Tjernström and Graversen, 2009), and are largely governed by the large scale atmospheric circulation (Stramler et al., 2011).

Specific humidity profiles over snow or sea ice covered surfaces in winter are nearly always characterized by specific humidity inversions (Vihma et al., 2014; Nygård et al., 2014; Naakka et al., 2018; Devasthale et al., 2011). Most of the wintertime specific humidity inversions in the lower troposphere are formed due to the radiative cooling and associated condensation, at the surface or the cloud top, and are thus tied to temperature inversions and saturated conditions (Curry, 1983; Naakka et al., 2018). Above 800 hPa, specific humidity inversions are more commonly independent from temperature inversions, directly
related to elevated advection of a moist air (Naakka et al., 2018).

In this study, we associate vertical profiles of temperature and humidity with atmospheric large scale circulation patterns with in the circumpolar Arctic (north of 65°N). The allocation of characteristic circulation types is based on Self-Organizing Maps





(SOMs). In the analyses, we have utilized radiosonde, in situ cloud cover and radiation observations, as well as ERA5
atmospheric reanalysis fields, for the winter months (DJF) of 2009–2018. Specifically, we (i) show how the shape of
temperature and specific humidity profiles vary with the large scale circulation, and assess the sensitivity of the profiles to the
large scale circulation in different regions within the Arctic, (ii) identify the vertical extent of the surface–atmosphere
interactions in different circulation types, and (iii) assess the roles of horizontal advection, clouds and surface energy budget,
all dependent on the large scale circulation, for the profile shapes. As a side product we, as part of the process, also evaluate
how well ERA5 reanalysis agrees with radiosonde observations on the response of the profiles to the large scale circulation;
this is a prerequisite for a meaningful study.

## 2 Data

### 2.1 ERA5 reanalysis

We used the latest global reanalysis ERA5 ((C3s), 2017; Hersbach et al., 2020), from the European Centre for Medium-Range
Weather Forecasts (ECMWF) for the Copernicus Climate Change Service (C3S). The spectral model resolution of ERA5 is
T639, while the horizontal resolution of the data used in this study is 0.25° x 0.25°. ERA5 has 137 vertical model levels and
the vertical spacing of model levels is denser near the surface. The lowest model level is located at 7–10 m over the surface,
there are 8 levels below ~200 m and 20 below 1 km. We used the lowest 58 levels to cover the whole air column from the
surface to approximately 300 hPa. In ERA5, a variety of atmospheric observations were assimilated into a numerical weather
prediction model, applying a four-dimensional variational data assimilation method. The assimilated observations include, e.g.
satellite, surface and radiosonde observations. However, even when a radiosonde observations was assimilated, ERA5
temperature and humidity profiles are not necessarily always similar to observations, as the comparisons in Sect. 4.4 show.
This is because impacts of observations on the reanalyses vary regionally due to observation and model uncertainty (Naakka
et al., 2019), and availability and quality of other observations.

Large scale circulation in the Arctic was clustered using SOM, based on ERA5 mean sea level pressure fields at 6 h interval
for the winter months (DJF) of 2009–2018. We then composite total cloud cover, total column cloud water (TCW), and surface
fluxes, as well as temperature and specific humidity fields on the model levels on each cluster. The vertical axis of model level
data was converted to geopotential heights, defined here as the distance from the surface. When comparing to radiosounding
profiles, temperature and specific humidity values of ERA5 were selected from the grid column closest to the radiosounding
station. Energy fluxes were extracted at hourly resolution from the associated ERA5 forecast fields and averaged over ±3 h
around the SOM time steps. The diurnal cycle is ignored since it is essentially absent in winter (Lemone et al., 2019).

Uncertainties related to circulation patterns represented by ERA5 are assumed to be small as atmospheric reanalyses are able
to accurately represent atmospheric pressure fields. Uncertainties in reanalysis temperature and specific humidity, and





especially in cloud cover, may be considerably larger and are evaluated as a part of this study. It is known from earlier studies that ERA5 performs well for the surface meteorology and water vapour profiles in winter over the Arctic sea ice and ice-free ocean (Graham et al., 2019; Renfrew et al., 2021). However, ERA5, similar to many other atmospheric reanalyses, has a near-surface winter warm bias, strongest at temperatures below -25°C (Wang et al., 2019; Graham et al., 2019). Graham et al.

(2019) concluded that reanalyses have difficulties in resolving vertical temperature gradients in very stable boundary layers. It is known from previous generation reanalyses that they generally capture spatial distribution and seasonal cycle of specific humidity inversion occurrence (Brunke et al., 2015; Naakka et al., 2018), however, specific humidity inversions are often weaker and less frequent compared to radiosonde observations (Naakka et al., 2018).

**2.2 Radiosoundings and cloud observations**

We use temperature and specific humidity profiles between the surface and 500 hPa from 32 Arctic radiosounding stations north of 65°N for the winter months (DJF) of 2009–2018. These were taken from the Integrated Global Radiosonde Archive version 2 (IGRA V2) (Durre et al., 2006; Durre and Yin, 2008; Ferreira et al., 2019), a global dataset of quality-assured radiosonde data. In IGRA V2, temperature and specific humidity data are provided at the mandatory pressure levels defined

by the World Meteorological Organization, and at significant levels where a sounding variable deviates a specified amount from linearity. The level of uncertainty of the data and the amount of vertical levels reported vary in time and space, as radiosonde types and data analysis procedures vary between countries. At some of the stations, there were also changes in the radiosonde type during the study period. Information on radiosonde types used at individual stations as well uncertainty characterization of radiosonde types is not easily obtainable. Temperature uncertainty of Russian radiosondes is likely a factor

of two larger than for more modern radiosonde types (Naakka et al., 2019; Ingleby, 2017), and their humidity measurements clearly suffer from cloud contamination above mid-level clouds (Ingleby, 2017). We have not adjusted for inhomogeneities due to changes in instrumentation or observation practices.

Utqiaġvik (formerly known as Barrow) and Sodankylä stations, studied in more detail in this paper, are part of GCOS

Reference Upper-Air Network (GRUAN; https://www.gruan.org/), established to provide high quality sounding observations with traceable uncertainty assessment and documented practices. During the study period, Vaisala RS92 radiosondes were used at the stations, replaced with Vaisala RS41 radiosondes in 2017 and 2018 at Sodankylä and Utqiaġvik, respectively. Uncertainty of Vaisala R92 radiosondes is estimated to be approximately 0.5°C for temperature and 5% for relative humidity (Dirksen et al., 2014) consistent with manufacturer information. Jensen et al. (2016) compared performance of Vaisala RS41

radiosonde against Vaisala RS92 radiosonde, using the twin-sounding method, concluding that differences were typically much smaller than the uncertainty estimates.



Cloud cover observations at 27 of the radiosounding stations were taken from the Integrated Surface Database (ISD) at 6 hour intervals. For remaining Arctic stations, the percentage of missing data was ≥30% and these stations were not included.

Incoming and outgoing longwave radiation, used to define radiatively clear and cloudy cases at Sodankylä and Utqiaġvik, was measured with Kipp&Zonen CG4 and CGR4 pyrgeometers, and with Eppley PIR, respectively.

## 3 Methods

### 3.1 Large scale circulation clustering

The identification and clustering of large scale circulation types was accomplished using the SOM method, a statistical

unsupervised learning method determining generalized patterns in data, developed by Kohonen (2001). SOM is a type of artificial neural network, hence a machine-learning approach, previously applied in climatology of synoptic patterns (Hewitson and Crane, 2002; Cassano et al., 2006), for addressing moisture transport (Mattingly et al., 2016; Nygård et al., 2019; Skific et al., 2009) and analysing temperature and humidity inversions in the Arctic (Yu et al., 2019). The ability of SOM to provide physically meaningful composites of the circulation patterns has been demonstrated in numerous studies (Hewitson and Crane,

2002; Cassano et al., 2006; Johnson et al., 2008). Here, the SOM method was applied to allocate 12 characteristic atmospheric winter (DJF) circulation types for 10 years (2009–2018) of mean sea-level pressure (MSLP) from the ERA5 reanalysis at 6 h interval. The input MSLP data were first re-gridded to an equal-area grid. The SOM method provides a two-dimensional array of gridded MSLP fields, having the characteristics and probability density of the input MSLP data. The size of the SOM array was subjectively selected to 3 x 4, to include the main features of regional circulation patterns and yet be sufficiently general

to enable conceptualization of the results.

The principle of the SOM method is described in detail in Kohonen (2001), Hewitson and Crane (2002), Cassano et al. (2006) and Gibson et al. (2017). Briefly, the SOM procedure includes initialization, training, evaluation and visualization. In this study, the random initialization scheme (Hewitson and Crane, 2002) was applied to create an initial SOM array, in which each

node had an associated reference vector with an equal dimension to the input MSLP data. During the training, each input data vector (i.e., each time step of MSLP data) was compared with the reference vector of each node and those reference vectors most similar to the input data vector were adjusted towards the input data vector. Training was continued until reference vectors converged. As a result, a SOM array of characteristic MSLP patterns was obtained. The SOM array nodes are organized so that most similar circulation patterns are located close to each other while the most dissimilar patterns are situated in the corners

of the array. As a final result each input MSLP field can be associated with the most similar node of the array. Here, we present composites of the MSLP fields from ERA5, not the reference vectors directly from the SOM (see Fig. 1 and Supplemental Fig. S1)

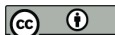



### 3.2 Analyses

The study covers the whole circumpolar Arctic north of 65°N. Radiosoundings, cloud cover observations and variables from
ERA5 were clustered according to their associated circulation types as allocated by the SOM analysis. Anomalies for each circulation type were calculated by subtracting the grid point medians within each circulation type from the corresponding median values for the whole time series. We used medians rather than mean values due to the often skewed non-Gaussian distributions of the analysed variables.

Statistical significance of anomalies within each circulation type was tested using a Monte Carlo approach. First, 3000 random subsets of a given field of ERA5 were constructed, having the same number of samples as each individual SOM circulation type. Then, anomalies in random subsets were compared to the anomalies in the SOM circulation types, by applying a two-tailed test at each grid point. An anomaly for a given SOM circulation type was assumed significant at the 99% level, if less than 0.5% of the anomalies in the random subsets were larger, or if negative smaller, than the anomaly for the SOM circulation
type. Similarly, a Monte Carlo approach was also applied to test the statistical significance of the anomalies of radiosonde and cloud cover observations for each circulation type.

In the analyses of cloud cover, clear sky and overcast conditions were identified from the surface observations (Sect. 2.2) and ERA5 reanalysis (Sect. 2.1). The sky was considered clear when 0-1/8 (<0.19 in ERA5) was covered by clouds, and overcast
when the coverage was 7-8/8 (>0.81 in ERA5).

Conditions at two stations, Utqiaġvik and Sodankylä, were studied in more detail. These were selected partly due to location and partly for having a large selection of high-quality observations. Estimates of cloud conditions for these two stations were made from radiation observations, considered more reliable since they directly represent the radiative impacts of clouds. The
use of data was maximized by identifying radiatively clear sky and cloudy (opaque) cases using the methodology of Stramler et al. (2011) based on the bimodal relative frequency distribution (RFD) of surface net longwave radiation flux (LWN). Similar to Sedlar and Tjernström (2019), the thresholds were LWN < –30 W m$^{-2}$ for the radiatively clear state and LWN > –20 W m$^{-2}$ for the radiatively cloudy state. Our own LWN analysis confirmed these thresholds as appropriate for both Utqiaġvik, and Sodankylä.

### 3.3 Profile metrics

To characterize thermodynamic profiles, we use the following simplified metrics: (i) maximum temperature and specific humidity in a profile; (ii) the difference between (i) and the corresponding surface values, and; (iii) the altitude to (i) above the surface. For simplicity, we will henceforth refer to (ii) as "bulk inversion strength", or just "inversion strength", while

recognizing that this metric will not distinguish between elevated inversions and surface inversions and will also miss elevated
inversions when the inversion top value is below the near surface value; it however works well for surface inversions.

## 4 Results

Large scale circulation has manifold ways to influence profiles of temperature and specific humidity in the Arctic. In addition
to direct impacts of horizontal transport (advection) of heat and moisture, largely driven by large scale circulation, atmospheric
circulation also influences cloud conditions and surface fluxes, which in turn strongly modify the temperature and humidity
profiles. After introducing the circulation patterns in Sect. 4.1, we first present the impacts of circulation on cloud conditions
(Sect. 4.2) and surface fluxes (Sect. 4.3) and then proceed to link circulation, cloud conditions and fluxes to profiles of
temperature and specific humidity in Sect. 4.4 and 4.5.

### 4.1 Atmospheric circulation patterns

As an outcome of the SOM analysis, 12 characteristic wintertime atmospheric MSLP patterns were identified and organized
in a 2-d array according to their similarities (Fig. 1). Corresponding anomalies of MSLP in these 12 circulation types, calculated
with respect to the DJF mean, are shown in Supplemental Fig. S1.

High pressure over northern Eurasia, and low pressures over the northern North Atlantic and Pacific are present in all
circulation types, whereas other features only occur in some of the circulation types with intensity, extent and location varying
between the 12 circulation types (Fig. 1). The main large-scale features appearing in the wintertime MSLP fields are: (i) an
extensive high pressure area over the Northern Eurasia, (ii) high pressure over Northern America, (iii) an area of high pressure
connecting the Eurasian and Northern American high pressure areas across the Arctic Ocean, previously referred to as the
Arctic Bridge (Nygård et al., 2019), (iv) high pressure over Greenland, (v) Icelandic low in the northern North Atlantic, and
(vi) Aleutian low in the Pacific sector.


The number of cases in each of the 12 circulation types varies between 147 (4% of the time) and 461 (13% of the time),
sufficient for statistical analyses. The most common circulation type, with an extensive high pressure in the Arctic, is type 10,
while the most persistent types (i.e., with the longest uninterrupted duration) are 1, 4, 9 and 10 (Fig. 1), in which the Icelandic
low has a relatively western location. The least frequent and least persistent circulation types are situated in the centre of the
array (types 5 and 8). The most distinct circulation types are situated in the corners of the SOM array (Supplemental Fig. S1);
in the following subsections we focus mostly on these patterns.



**Figure 1: Mean sea level pressure (MSLP) averaged over the cases belonging to each of the circulation types (SOM nodes) in winters (DJF) of 2009–2018. The values in the parentheses indicate the number of cases belonging to the circulation type and the median persistence of the circulation type.**



## 4.2 Cloud conditions linked to circulation patterns

First, we shortly present the median cloud conditions in the circumpolar Arctic to facilitate interpretation of cloud anomalies
related to different circulation types. The median cloud cover is rather uniformly high in the winter Arctic, mostly 7–8/8 both
in ERA5 and cloud cover observations (Fig. 2a); note that cloud cover observations in the polar night may be biased low due
to light conditions (Hahn et al., 1995). Because the wintertime cloud fraction distribution in the Arctic is bi-modal, peaking at
clear sky and overcast conditions, we also separately show occurrences of the clear sky (cloud fraction $\leq 1/8$) and overcast
conditions (cloud fraction $\geq 7/8$) (Fig. 2c–d). Overcast conditions occur more than 50% of the time in most of the circumpolar
Arctic. Clear sky conditions are rare; the Canadian Arctic Archipelago and the northern coast of Greenland being exceptions.
The median TCW, as analysed from ERA5, displays more spatial variations with median values higher than 0.05 kg m$^{-2}$ only
in some parts of the ice-free ocean areas, mainly along the ice edge and over land in Scandinavia and the western part of Russia
(Fig. 2b). In these regions, a substantial part (mostly 30–70%) of the TCW is liquid water (Supplemental Fig. S2), which is
more efficiently suppressing surface radiative cooling than cloud ice. Over Siberia, the central Arctic Ocean, North America
and Greenland, the median TCW is low, less than 0.01 kg m$^{-2}$. In these areas, the clouds mostly consist of ice; the variations
are also mostly due to cloud ice variations (Supplemental Fig. S2), hence limiting radiative impacts.

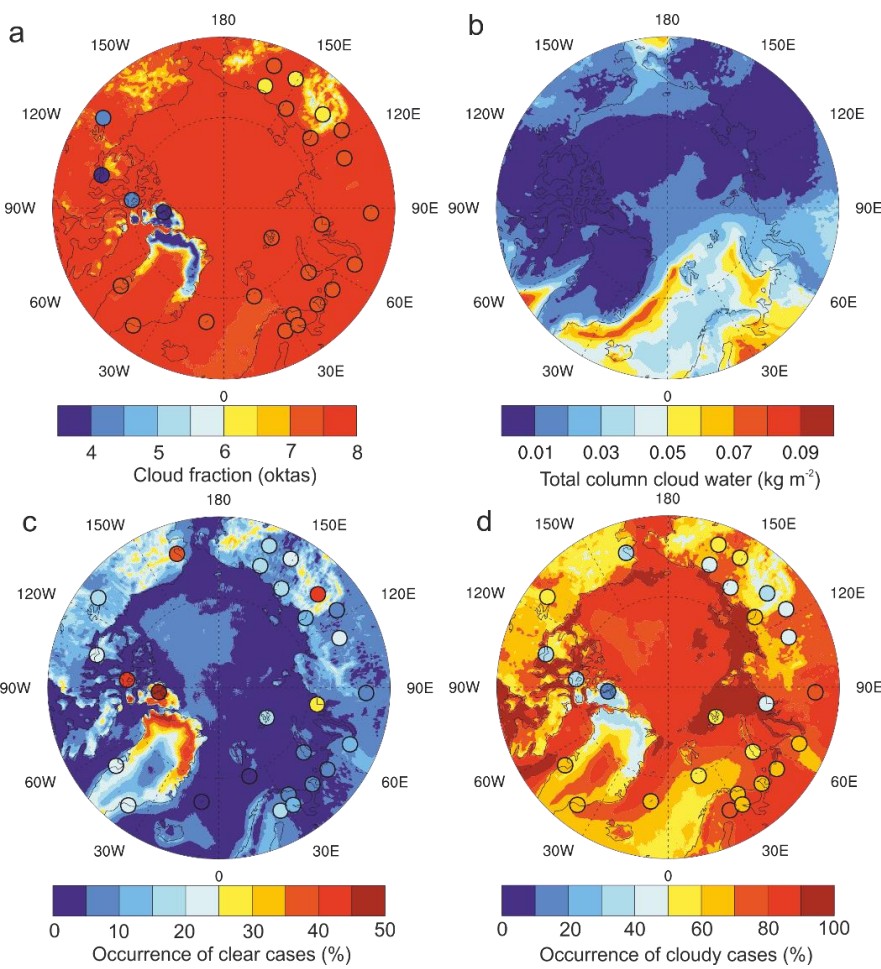

**Figure 2: (a) Median cloud fraction, (b) median total column (liquid and frozen) cloud water, (c) occurrence of clear (cloud fraction**
**≤ 1/8) cases and (d) occurrence of overcast (cloud fraction ≥ 7/8) cases in December–February, 2009–2018. The colours inside the**
**circles in (a), (b) and (c) show synoptic cloud observations at the radiosounding stations, whereas the fields in (a)-(d) are based on**
**ERA5.**

We demonstrate the general relationships between the near-surface circulation and clouds by showing results for the four
corner circulation types (Fig. 3). In circulation type 1, characterized by anomalous high pressure over Alaska and lower than
average MSLP elsewhere in the Arctic (Fig. 3a and Supplemental Fig. S1), the TCW and occurrence of overcast conditions
are both anomalously high around the Bering Strait and in the northern North Atlantic (Fig. 3e and m). These positive
anomalies are linked to anomalous northward meridional transport from the northern North Atlantic and North Pacific oceans.
Negative anomalies in TCW, especially liquid, and in occurrence of overcast cases (Fig. 3e and i) are found in the north-
western parts of Eurasia, linked to a lack of influence from the North Atlantic and weak transports of heat and moisture from
open oceans. In circulation type 3, an anomalous low pressure is centred over the Barents Sea and MSLP is lower than average
in the whole Eurasia (Fig. 3b and Supplemental Fig. S1). This circulation type is associated with substantial positive anomalies



in TCW and overcast occurrences in the sector between 30°E and 90°E, influenced by a stronger than average westerly flow and advection of warm and moist air from the North Atlantic, linked to the low pressure (Fig. 3f and n). Especially between 60°E and 90°E, the fraction of liquid cloud water is clearly anomalously high (Supplemental Fig. S2).

Circulation type 10 has anomalously high MSLP in most of the Arctic, especially over the Arctic Ocean and Greenland (Fig. 3c and Supplemental Fig. S1). This type is associated with a generally lower TCW and increased clear skies (Fig. 3g and k). However, anomalies in cloudiness in the central Arctic cause only weak anomalies in the net longwave radiation (not shown) because TWC is generally low and the anomalies are mostly due to cloud ice. In circulation type 12, there is an extensive high pressure area over the northern Eurasia, whereas the high pressure connecting Eurasia and Northern America across the Arctic Ocean (Arctic Bridge) is absent (Fig. 3d and Supplemental Fig. S1). In this circulation type, relatively moist and warm air from the North Atlantic flows eastward over the Barents, Kara and Laptev Seas, and provides moisture facilitating anomalously high amount of TCW in these, mostly sea ice covered, regions. Advection of warm air also increases the fraction of liquid water in clouds here (Supplemental Fig. S2). In contrast, the land area in the north-western parts of Eurasia has anomalously low TCW and more frequent clear skies.

While clear skies are slightly more common in observations than in ERA5, reanalysis and observations mostly agree. Overcast conditions are up to 20 percentage points more common in ERA5 than in observations at many stations. Importantly, ERA5 captures the variability of cloud cover between the circulation types, although the differences at a few stations to the observations are substantial (Fig. 3). For the purpose of this study, where we mainly focus on the clear and cloudy conditions, we conclude that ERA5 adequately represents the cloud anomalies associated with different circulation types in the wintertime Arctic.



**Figure 3: Mean sea level pressure (a–d), anomaly of total column (liquid and frozen) cloud water (e–h), anomaly of occurrence of clear skies (cloud fraction ≤ 1/8) (i–l), and occurrence of overcast conditions (cloud fraction ≥ 7/8) (m–p) in four SOM circulation types (indicated on the top of the column) during December–February 2009–2018. The colours inside the circles in (i–p) show synoptic cloud observations at the radiosounding stations, whereas the fields in (a)-(p) are based on ERA5. The dotted shadings (ERA5) and black circles (synoptic cloud observations) show the anomalies that are different from zero at the 99% significance level.**





### 4.3 Surface fluxes linked to circulation patterns

For all surface energy fluxes shown in Fig. 4, we present median values instead of anomalies, to clearly display their directions. Since solar radiation is absent and the surface albedo is very high a large part of winter, the surface radiation budget is dominated by net longwave radiation; the distribution is bimodal (Stramler et al., 2011) while the median is negative (Fig.

4m–p). Consequently, radiative surface cooling dominates over the circumpolar Arctic, especially over the Arctic Ocean. Surface net radiation variations between different circulation types are tightly connected to distributions of TCW; net radiation is less negative when TCW is higher (compare Figs. 3e–h and 4m–p) and *vice versa*.

On snow and ice-covered surfaces, the turbulent heat fluxes compensate for the net radiation at the surface, contributing to

shape the temperature and humidity profiles by setting lower boundary conditions. Therefore, reduced surface radiative cooling immediately leads to surface warming (or melting if the temperature is high enough). The turbulent heat fluxes are small and commonly directed downward; condensation/sublimation of water at the surface is common. Any remaining residual in the energy budget is presumably closed by conduction of heat through the ice. This is not available from atmospheric reanalysis since it requires accurate modelling of ice thickness. Latent heat flux is very small for all circulation types since specific

humidity is small to begin with, and the sensible heat flux is mainly responsible for balancing radiative cooling. Hence, over snow and ice, spatial patterns of sensible heat flux correspond to the patterns of net radiation, in turn closely linked to cloud distributions controlled by large scale circulation.

Over open water, on the other hand, the large heat capacity of ocean surface layer prevents rapid changes in the sea surface

temperature linked to changes in radiative cooling. Relatively warm open water surface is nearly always a source of atmospheric heat and moisture. Over open ocean, strong upward turbulent heat fluxes lead to efficient vertical mixing, and formation of a mixed layer in the lower atmosphere. Sensible and latent heat fluxes over the open ocean have clear variations linked to the large scale circulation; the largest sensible and latent heat fluxes occur during southward flows in the northern North Atlantic (circulation types 3 and 10) and the smallest during northward flows in the northern North Atlantic (circulation

types 1 and 12) (Fig. 4e–l).



**Figure 4: Mean sea level pressure (a–d), sensible heat flux (e–h), latent heat flux (i–l) and net radiation at the surface (m–p) in four SOM circulation types (indicated on the top of the column) during December–February 2009–2018 based on ERA5. For sensible and latent heat fluxes, positive direction is defined to be upwards, and for the net radiation downwards. Note the nonlinear scale for the latent heat flux.**





## 4.4 Characteristics of temperature and humidity profiles linked to circulation patterns

Within the Arctic the characteristics of the vertical thermodynamic structure is to a first order controlled by the surface type,
whereas temporal variations at a certain location are mostly controlled by the synoptic weather conditions. Figures 5–7 show
profile metrics (defined in Sect. 3.3) for both ERA5 and observations from selected sounding stations. There is in general a
good agreement between the two, not unexpectedly since most of these soundings were assimilated into the reanalysis.

Unsurprisingly, temperature and specific humidity maxima, characterizing the air mass properties, are largest over open ocean
(Fig. 5a and d) where they are also typically located at the lowest model level (Fig. 5c and f). This is because the open water
surface is a source of both heat and moisture (see Sect.4.3). Over land or sea-ice, the maxima are typically found at a higher
altitude, linked to the cooling of the atmosphere by the surface (see Sect. 4.3). Both temperature and specific humidity maxima
are typically found below 1500 m (Fig. 5c and f). In most of the Arctic, the specific humidity maximum is located slightly
higher than that for temperature; the largest differences between the two are found over the Beaufort Sea and Central Siberia.
Bulk inversion strengths are largest over the continents both for temperature and specific humidity (Fig. 5b and e); the strengths
for temperature exceed 8°C over most of the North American and Russian Arctic, while the strengths for specific humidity
exceed 0.5 g kg$^{-1}$.

In circulation type 1, anomalously warm and moist air masses are found around the Bering Strait (Figs. 6a and e, 7a and e),
associated with increased TCW (Fig. 3e), reduced radiative cooling (Fig. 4m), and weak temperature and specific humidity
inversions (Figs. 6i and 7i). The vertical maxima of temperature and specific humidity are at anomalously low levels (Figs.
6m and 7m). Elsewhere in the Arctic anomalies are weaker and at some locations ERA5 and radiosoundings disagree on sign:
Based on ERA5, temperature and specific humidity inversions are anomalously weak and the vertical maxima of temperature
and humidity are located at an anomalously low altitude in most of the Arctic, but radiosonde profiles indicate more commonly
positive bulk inversion strength anomalies and higher altitudes of the vertical maxima.




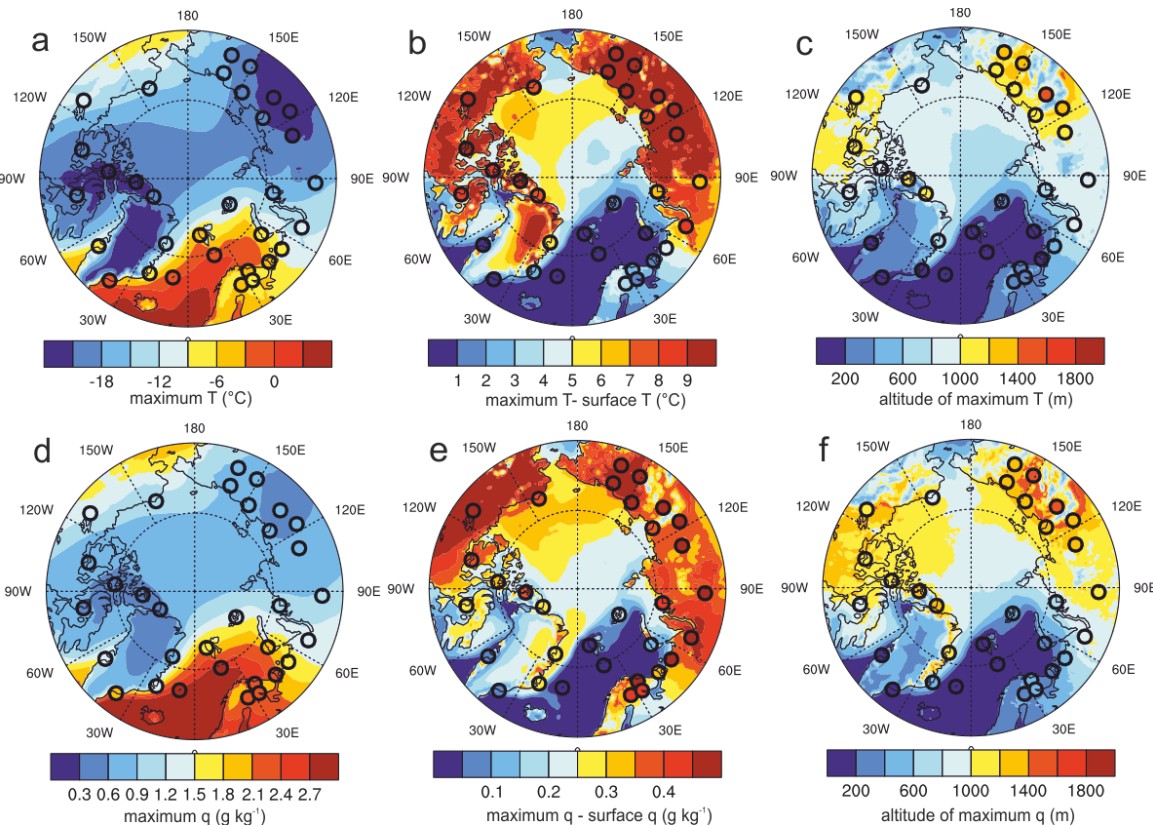

**Figure 5: Medians of (a) vertical maximum T, (b) vertical maximum T–surface T (bulk inversion strength), (c) altitude of vertical maximum T, (d) vertical maximum q, (e) vertical maximum q–surface q (bulk inversion strength), and (f) altitude of vertical maximum q, in December–February 2009–2018. The colours inside the circles show values based on radiosonde observations, whereas the fields are based on ERA5.**

When circulation type 3 dominates, clear linkages between atmospheric circulation and thermodynamic profiles are indicated; here ERA5 and radiosonde profiles agree. Anomalously warm and moist air is found over the Eurasian land, due to transport of heat and moisture related to the low pressure (Figs. 6b and f, 7b and f). The positive Eurasian anomaly in TCW and occurrence of overcast cases between 30°E and 90°E (Fig. 3f and n) reduce radiative cooling at the surface (Fig. 4n) and contribute to the erosion of temperature and specific humidity inversions in this sector (Figs. 6j and 7j). Consequently, temperature and specific humidity are at their maximum at the lowest model level (Figs. 6n and 7n). However, warm and moist air reaches all the way to 120°E, but its effects on the thermodynamic profiles are rather different in the central Russian Arctic (between 90°E and 120°E) compared to farther west. In the central Russian Arctic, the TCW is close to the median with slightly increased radiative cooling at the surface. As a consequence, inversions are anomalously strong (Figs. 6j and 7j). It is also noteworthy that the anomalously cold and dry air over the northern North Atlantic and Bering Strait, connected to strong upward sensible and latent heat fluxes over the open ocean (Fig. 4f and j), has a very small effect on the thermodynamic profile





metrics, because our metrics emphasize the inversion layers. However, this cold and dry air is associated with weak negative anomalies in bulk inversion strength of temperature and humidity over the sea ice.

Circulation type 10 leads to isolation of the high Arctic from poleward transport of air from the northern North Atlantic. During this circulation type, temperature inversions in the central Arctic Ocean are anomalously strong (Fig. 6k), linked to radiative

cooling at the surface and probably also to subsidence. The vertical maxima of temperature are found several hundreds of meters higher than the median over the Eurasian Arctic and the Laptev, East Siberian and Chukchi Seas (Fig. 6o). Temperature inversions in the central Russian Arctic, where clear sky conditions are anomalously frequent, are stronger than the median. However, westward from 60°E, temperature inversions are anomalously weak, associated with more frequent overcast conditions and hence reduced radiative cooling at the surface, even if the amount of cloud water is low. Hence, the temperature

inversions in the cold sector over the Eurasian continent are strongly affected by cloud cover. The cold air limits specific humidity in the sector from 30°W anticlockwise to 120°E (Fig. 6g), and this, possibly together with a drying effect of subsidence, limits the inversion strength of specific humidity (Fig. 7k). However, over the continents here, the vertical maximum of specific humidity is located 100–300 m higher than the median (Fig. 7o), probably due to a long residence time of air over the continent, which has allowed the growth of a persistent inversion layer. Warm and moist air over Greenland

and Baffin Bay and over the land in Pacific sector, from 150°W to 130°E, does not have any unambiguous impacts on temperature and specific humidity profiles. Generally, ERA5 and radiosoundings agree on the main regional features of the profile metrics, but not in all regional details.

For circulation type 12, the largest bulk inversion strength anomalies of temperature and specific humidity are found over

Scandinavia and Western Russia (20°E to 90°E) (Figs. 6l and 7l), where the air is cold and dry and TCW is low (Figs. 6h and 7h). Likely, the air has spent a relatively long time over the cold continent, facilitating uninterrupted surface cooling over prolonged time. In this region, temperature inversions are 3–4°C stronger while specific humidity inversions are 0.05–0.2 g kg$^{-1}$ stronger than their medians. The high pressure over the northern Eurasia steers warm and moist North Atlantic air over the Barents, Kara and Laptev Seas, where inversions are partly eroded. This warm and moist air reaches in over the land in

central Siberia, where temperature inversions are notably weakened, probably linked to the radiative impacts of clouds. The cold and dry air in the Pacific sector is linked to anomalously strong temperature inversions over land, but specific humidity inversions are anomalously weak in this generally dry air. Over the sea-ice covered Arctic Ocean with cold and dry air, temperature and specific humidity inversions are weak, linked to the mainly upward directed sensible heat flux over the sea ice. ERA5 and radiosoundings mostly agree on the main regional features of the profile metrics in this circulation type.


©c Author(s) 2021. CC BY 4.0 License.



**Figure 6: Mean sea level pressure (a–d), anomaly of vertical maximum T (e–h), anomaly of vertical maximum T–surface T (bulk inversion strength) (i–l), anomaly of altitude of vertical maximum T (m–p) in four SOM circulation types (indicated on the top of the column) in December–February 2009–2018. The colours inside the circles show values based on radiosonde observations, whereas the fields are based on ERA5. The dotted shadings (ERA5) and black circles (radiosonde observations) show the anomalies that are different from zero at the 99% significance level.**



**Figure 7: Mean sea level pressure (a–d), anomaly of vertical maximum q (e–h), anomaly of vertical maximum q–surface q (bulk inversion strength) (i–l), anomaly of altitude of vertical maximum q (m–p) in four SOM circulation types (indicated on the top of the column) in December–February 2009–2018. The colours inside the circles show values based on radiosonde observations, whereas the fields are based on ERA5. The dotted shadings (ERA5) and black circles (radiosonde observations) show the anomalies that are different from zero at the 99% significance level.**



### 4.5 Vertical profiles at Utqiaġvik and Sodankylä

**4.5.1 Utqiaġvik (formerly Barrow)**

Figure 8 shows vertical profiles and gradients of temperature and specific humidity for Utqiaġvik in Alaska (see location in Fig. 8 a–d) for circulation types 1, 3, 9, and 10, representing distinct large-scale flow patterns at this location. Both radiatively clear and cloudy conditions, based on the radiation observations, are frequent for all of these four circulation types; percentages are 33–39% and 51–55% for cloudy and clear cases, respectively. By investigating profiles separately in radiatively cloudy and clear cases, we are better able to reveal the relative roles of directs impacts of the circulation and local impacts due to cloud conditions.

The radiosounding results for Utqiaġvik (Fig. 8) show that the radiatively cloudy cases are generally warmer and moister in the lowest 3000 m than the radiatively clear cases, for all four circulation types. This is presumably partly due to different properties of air masses leading to cloudy and clear conditions (especially circulation type 10) and partly due to radiative effects of clouds, warming the profile below the cloud layer, especially in circulation type 1. Differences in temperature gradients between cloudy and clear conditions are mostly seen in the lowest 400 m, where temperature gradients in cloudy profiles are generally smaller, sometimes even negative. However, the shape of the temperature profiles varies more between circulation types than between different cloud conditions within a certain circulation type, suggesting a major role of large-scale circulation for shaping the thermodynamic profiles. Furthermore, the cloudy temperature profiles vary more between circulation types than those for radiatively clear situations, suggesting that also properties of clouds depend on large scale circulation. Even if median near-surface temperature is almost the same across the large scale circulation types, vertical gradients and altitude of temperature maxima vary largely. A temperature inversion is present in the median profiles for all the four circulation types (Fig. 8). Surface-based temperature inversions are eroded in low pressure situations (circulation type 9), leading to strong elevated temperature and humidity inversions, with the largest positive temperature and specific humidity gradients between 300m and 800 m. On the other hand, high pressure (circulation types 1, 3, and 10) is associated with strong vertical gradients of temperature below 500 m, whereas specific humidity gradients are more equally distributed within the lowest 1500 m; circulation type 3 with strong near-surface humidity gradients is an exception. In general, the differences in the profile shape between different large scale circulation types are mostly seen below 1500 m.

**Figure 8: Median (solid lines), 10% percentile (dashed lines), 90% percentile (dashed lines) of temperature profiles (e–h), specific humidity profiles (i–l), vertical temperature gradient profiles (m–p), and vertical specific humidity gradient profiles (q–t) at Utqiaġvik for four atmospheric circulation types (a–d) in winter during 2009–2018. The profiles are based on radiosounding observations. Red lines show radiatively clear profiles, blue lines radiatively cloudy profiles. The caption is shown in (p). The location of Utqiaġvik is marked with a circle in (a–d). Note the different vertical scale in m–t, compared to e–l. Gradients are defined positive, when the values increase upwards. Note the different vertical axis in m–t compared to e–l.**





### 4.5.2 Sodankylä

For Sodankylä in Northern Finland (see location in Fig. 9 a–d), we explore circulation types 12, 1, 4, 5 (Fig. 9). The ordering of circulation types in Fig. 9 is arranged according to the location of the Atlantic low, shifting from west to east. During the four circulation types, percentages of radiatively cloudy and clear cases are 65–82% and 11–27%, respectively.

Variability in the observed temperature and specific humidity profile shapes in Sodankylä is mostly seen below the altitude of
500 m, thus in a 1000 m shallower layer compared to Utqiaġvik. Differences in temperature profiles between the cloudy and clear cases are also smaller than in Utqiaġvik, although the absolute differences in specific humidity profiles between the cloudy and clear cases are larger in Sodankylä, due to generally higher temperatures. Cloudy cases are generally moister than the clear cases, suggesting that cloudy and clear conditions in Sodankylä are associated with different properties of air masses, similar to Utqiaġvik. Strong, positive temperature gradients are only seen in clear conditions, whereas positive specific
humidity gradients are found in both clear and cloudy conditions. Generally, variability in wintertime temperature profiles is largely related to the circulation; whether Sodankylä is under influence of Eurasian high pressure conditions, with weak flow (circulation types 12 and 1), or low pressure conditions, with an air flow from the Northern North Atlantic and southern latitudes (circulation types 4 and 5). High pressure conditions are linked to colder and drier vertical profiles, in particular in the lowest 500 m, whereas low pressure conditions erode most of the surface-based temperature and specific humidity
inversions.





Figure 9: Median (solid lines), 10% percentile (dashed line), 90% percentile (dashed line) of temperature profiles (e–h), specific humidity profiles (i–l), vertical temperature gradient profiles (m–p), and vertical specific humidity gradient profiles (q–t) at Sodankylä for four atmospheric circulation types (a–d) in winter during 2009–2018. The profiles are based on radiosounding observations. Red lines show radiatively clear profiles, blue lines radiatively cloudy profiles. The location of Sodankylä is marked with a circle in (a–d). Note the different vertical scale in m–t, compared to e–l. Gradients are defined positive, when the values increase upwards. Note the different vertical axis in m–t compared to e–l.



## 5 Discussion

Our results indicate that there are notable regional differences in how atmospheric circulation affects the temperature and specific humidity profiles, especially inversion layers. These regional differences are tightly linked to air mass transformation from warmer and moister mid-latitude air to colder and drier polar air by longwave radiative cooling. Based on the linkages between large scale circulation and vertical profiles of temperature and humidity, we divide the study area into five regions with specific characteristics:


(1) Over the Arctic sea ice, the maximum vertical air temperature varies largely between the circulation types. These variations are connected to different physics: horizontal advection, subsidence, latent heating, or periods with long residence time and uninterrupted radiative cooling, as earlier shown by Messori et al. (2018) and Papritz (2020). Hence the circulation patterns sometimes have a direct effect, like with warm air intrusions, and sometimes the effects are due to local physics but enabled

by a circulation pattern. Over the central Arctic Ocean, the strongest temperature inversions are associated with the strongest high pressures, in which TCW is low and subsidence warms and dries the air, while surface radiation cools the air from below, hence, the warmest air does not reach all the way to the surface. In these conditions, the air is dry, due to the low saturation temperature and also a probable drying effect of the subsidence, which mostly keep the specific humidity inversions weak. In other words, our results indicate that the coldest air masses over the Arctic sea ice are not connected to the strongest temperature

inversions in the region. This is probably due to the heat flux through the sea ice, causing the temperature inversions to be weaker over the sea ice compared to land surface (Serreze et al., 1992). We also find that over sea ice, the maximum temperature is typically found between 500 and 1000 m, but the maximum specific humidity between 600 and 1200 m. For both, the altitude of the maxima varies roughly ±300 m between the circulation types.

(2) Over the open ocean, temperature and specific humidity maxima vary largely with large scale circulation. The relatively warm surface and consequently mostly upward directed heat fluxes keep the maxima nearly always near the surface; however, warm air advection may lead to stable stratification or even a shallow temperature inversion. Our metrics, focused on bulk inversion strength and vertical maxima, do not reveal the characteristic of the well-mixed layer over open ocean, although this layer is known to be influenced by atmospheric circulation and its effects on the surface fluxes. For example, certain flow

patterns linked to pressure dipoles lead to cold air outbreaks over open water near the sea ice margin (Kolstad et al., 2009; Pithan et al., 2018). These cold air outbreaks, associated with very large turbulent surface fluxes, lead to a well-mixed boundary layer, initially very shallow but growing with fetch over the open water (Brümmer, 1996). These flows are also associated with some of the largest turbulent surface energy fluxes recorded on Earth, leading to rapid warming and moistening of the air.





(3) Over Eurasian land west of 90°E, a large variability in the temperature and humidity profiles is seen, suggesting that the profiles are sensitive to large scale circulation. This region is occasionally influenced by low pressure systems facilitating large transport of heat and moisture from the North Atlantic. Under those conditions, the temperature and specific humidity inversions are typically eroded and the vertical maxima of temperature and specific humidity are found at the surface. This erosion of temperature surface inversions has earlier been pointed out by Serreze et al. (1992). Occasionally, this region is

affected by the extensive high pressure area over Eurasia, associated with cold and dry air and low TCW due to a long residence time of the air over the cold continent. Our results indicate that in high pressure conditions, temperature and specific humidity inversions are much stronger, but still weaker than over the most of the Arctic land area; a similar spatial distribution has been earlier reported by Zhang et al. (2011) and Naakka et al. (2018). The maximum temperature in this region is typically found between 200 m and 1000 m altitude, and some circulation types cause as large as ±400 m anomalies to this altitude.


(4) Over Eurasian land east of 90°E, well-developed polar air with a long residence time over the cold continent dominates. Here, temperature and specific humidity inversions are strong, as earlier shown by Serreze et al. (1992), Zhang et al. (2011) and Naakka et al. (2018). Our results indicate that variability of bulk inversion strengths of temperature is related to the strength of the high pressure and the amount of cloud water, which is mostly ice. Specific humidity inversions are weak when the air

is dry, presumably due to the low temperature and any drying effect of subsidence. Interestingly, warm and moist air originating from the northern North Atlantic is occasionally able to reach this region, with an extensive high pressure over the northern Eurasia and absence of an Arctic Bridge (high pressure) across the Arctic Ocean (circulation type 12). Hence, the break-up of the Arctic bridge not only allows for efficient transport of moist and warm Northern Atlantic air masses into the central Arctic, as earlier indicated by Nygård et al. (2019), but also enables these air masses to occasionally reach all the way into eastern

Eurasia. These warm and moist air masses make the temperature and specific humidity inversions notably weaker, but are typically unable to erode them completely. Temperature and specific humidity maxima are mostly found above 1000 m; the different circulation types typically cause variations of a few hundred meters.

(5) Over North America and Greenland, impacts of the large scale circulation on temperature and specific humidity profiles are less systematic than for other regions. This is probably due to mostly modest MSLP variations compared to the rest of the

Arctic, expect for in Alaska. Over Alaska, heat advection related to the Aleutian low and a high pressure centred east of Alaska, and anomalous downward longwave radiation, due to clouds, are known to cause warm extremes (Cassano et al., 2016), while anticyclonic circulation reduces cloudiness and leads to stagnation of the cold air and strong temperature inversions (Sulikowska et al., 2019). Especially the latter is seen also in our results for Utqiaġvik. For Greenland, Shahi et al. (2020) reported that surface-based temperature inversions are stronger in the northern and eastern parts of Greenland, suggesting that

this spatial distribution is due to the dominance of the Greenland high in the north. In our results, we also see this spatial distribution within Greenland, but its temporal variations are not clearly associated with any certain circulation pattern.



## 6 Conclusions

We have explored the impact of large scale circulation on the vertical structure and clouds in the circumpolar Arctic region using ERA5 in combination with Self Organizing Maps and observations, especially radiosoundings. We found that radiosonde
observations and ERA5 reanalysis mostly agree on the response of the temperature and specific humidity profiles to large scale circulation. We conclude that:

- Large scale circulation determines the origin and pathway of an air mass. The air mass experiences transformation along the pathway, e.g. due to radiative cooling and phase changes of water. Locally, effects of large scale circulation
are manifested as heat and moisture advection, which have first order effects on temperature and specific humidity profiles and the turbulent surface fluxes. The advection may also lead to cloud formation, which will cause second order effects on the fluxes and profiles, e.g. by shifting the strongest radiative cooling from the surface to the cloud top and forming an elevated temperature inversion. The second order effects have a major role especially over snow and ice-covered surfaces, where temporal variations in spatial patterns of surface sensible heat flux correspond to
patterns of net radiation, which are tightly linked to cloud distributions largely controlled by large scale circulation.

- The most sensitive regions to wintertime large scale circulation are the Eurasian land area west of 90°E and the Arctic Ocean sea ice region, whereas impacts over the North American Arctic and Greenland are weaker and more ambiguous. Eurasian land between 90°E and 140°E is typically not influenced by transport of oceanic air; however,
a circulation pattern, with an extensive high pressure over the northern Eurasia without a continuous high pressure bridge across the Arctic Ocean, enables warm and moist air masses originating from the northern North Atlantic to cross over the Barents, Kara and Laptev Seas and reach this Eurasian sector. Along this pathway, temperature and specific humidity inversions are partly eroded over the seas and substantially weakened over the land, due to the heat and moisture advection and radiative impacts of clouds.


- The altitudes of maximum temperature and specific humidity, and their variability between the circulation types, are good indicators of the depth of the layer impacted by surface–atmosphere processes interacting with the large-scale situation, except over the open ocean. This layer is deepest (~1500 m) over Eurasia, east of 90°E, and Northern America, and shallowest (~800 m) over western Eurasia, west of 60°E. Different circulation types typically cause
variations of a few hundred meters to the altitudes, except over Greenland, where variations are smaller.

Hence, the influence of different large scale flows on the Arctic must be viewed as a progressing set of processes. The primary reason is the advection – or lack thereof – of air masses into the Arctic, but then follows transformation of the air through various physical processes, differently for different circulation types but always facilitated by the circulation. Therefore, an





understanding of conditions in a certain region, how it varies and may change, must incorporate understanding of both the

large scale circulation and the ensuing physical processes affecting the air mass transformation.

**Code availability**

Code for the analysis is available upon request.

**Data availability**

ERA5 data are freely available from the Copernicus Climate Data Store (https://cds.climate.copernicus.eu), IGRA data from

the National Climatic Data Center (https://www1.ncdc.noaa.gov/pub/data/igra/), and cloud cover data from the Integrated

Surface Database (https://www.ncdc.noaa.gov/isd). The radiation data from Utqiaġvik is provided by the Department of

Energy's Atmospheric Radiation Measurement (ARM) program (https://www.arm.gov/), and from Sodankylä by the Finnish

Meteorological Institute (https://en.ilmatieteenlaitos.fi/download-observations).

**Acknowledgements**

This work was funded by the Academy of Finland via project TODAy (308441). TNY is also grateful to the International

Meteorological Institute (IMI) at Stockholm University. MT is funded by the Swedish Research Council (Grant 2016-03807)

and the Knut and Alice Wallenberg Foundation (KAW 2016-0024). The authors thank the data providers, Markku Kangas for

assistance with the radiation data from Sodankylä, and P. B. Gibson and P. Uotila for providing the codes for SOM analysis.

**Author contributions**

TNY and MT designed the study. TNY carried out the analysis in collaboration with TNA. All authors contributed to

interpretation of the results and writing.

**Competing interests**

The authors declare that they have no conflict of interest.



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
