# Peer review of "Winter thermodynamic vertical structure in the Arctic atmosphere linked to large scale circulation"

_Weather and Climate Dynamics, 2021_

## Referee Comment (RC1)

**Review of *Weather and Climate Dynamics* manuscript #wcd-2021-41**

Winter thermodynamic vertical structure in the Arctic atmosphere linked to large scale circulation

by T. Nygård, M. Tjernstöm, and T. Naakka

**Overview**

This manuscript describes an analysis of Arctic winter atmospheric properties associated with various characteristic large-scale circulation regimes identified using a clustering tool called self-organizing maps. The authors use reanalysis output, station data, and radiosonde profiles to investigate differences in vertical temperature and moisture profiles, cloud amount and liquid water content, inversion strength and height, and surface energy fluxes associated with varying circulation and advection regimes based on surface pressure. The manuscript is well written, figures are clear, and the analysis is thorough, but the results and conclusions mostly confirm what is already known and do not provide substantial new understanding. In general, they find that conditions in which a warm, moist flow from open-ocean areas penetrates the Arctic are associated with increased cloudiness, increased liquid water content, increased downwelling longwave radiation, and weaker inversion strengths. Circulation regimes that cause flows from ice-covered or land areas tend to be drier with lower cloud amounts, stronger inversions, and smaller downwelling longwave fluxes. The only surprising finding to me was the higher altitude of maximum temperature and specific humidity values in conditions of stronger inversions; I would have expected the opposite. Nevertheless, the study provides a very instructive summary of a complex environment that would likely be useful for educational purposes and for ongoing studies to understand processes associated with the rapid changes that are underway in the Arctic. For these applications, I would support publication. The study would have been much more interesting and enlightening if the authors had explored differences in atmosphere/regime linkages during the "cold Arctic" period (say, 1979-1996) versus the recent decades presented in this manuscript (2009-2018).

**Specific comments**

1. Section 4.2: It's unclear whether Fig. 2 is based on all days in the data set or a subset.

2. In Figure S2, why is the high fraction of liquid water collocated with high pressure in types 1, 3, and 10 but much less so in type 12?

3. Line 224: change shortly to briefly

4. 203-204: In my experience, using anomaly fields to create the SOM eliminate the problem of all types exhibiting some of the same features. Anomaly fields accentuate differences among types and often assist in interpreting other fields mapped to the SOM. It would be interesting to see if results changed if anomalies had been used to create the SOM, and how different the types would look compared to Fig. S1.

5. 224: change shortly to briefly

6. 268-273: Is ERA5 able to capture realistic differences in cloud phase? This seems an important question given the sensitivity of surface fluxes to this variable.

7. Section 4.3: I'm surprised that more attention has not been given to precipitable water (total column water vapor), as it has a large impact on downwelling longwave fluxes in winter, especially when clouds are absent.

8. 296: This statement may not hold as the sea ice becomes much thinner and broken.

9. Figures 4 e-p: It's very difficult to see important differences among types. I suggest instead plotting differences from median values to make differences more conspicuous. The color scale used for m-p is also not optimal for displaying differences.

10. 322: I suggest adding "and resulting inversion strength" after "by the surface".

11. 323-325: How accurate are the elevations of T and q maxima? Is the difference between the heights statistically significant?

12. 325-327: I suggest providing the standard deviation to give a sense of variability in these values.

13. 332-335: It's surprising that ERA5 and raobs differ given that the raobs are assimilated into the reanalysis. Perhaps type 1 collects a relatively wide variety of circulation patterns, which is typical for corner nodes of the SOM.

14. 360-361: Can you offer an explanation for this finding?

15. 370: "does not have any unambiguous impacts" is a convoluted statement – can you reword to clarify?

16. 412: How are temperature gradients defined/measured?

17. 419: change largely to greatly

18. 435: Please explain why a different set of SOM types from the rest of the analysis are used for this location.

19. 440-441: Could this difference just be due to the different types selected for analysis?

20. 470-471: Perhaps the duration of the high pressure is also a factor.

---

## Author Response (AR1)

*Dear Editor and Reviewers,*

*We highly appreciate all the constructive comments and suggestions we received in order to improve our manuscript and we have tried our best to take them into account in this revised manuscript. Our response is written in italics after each comment.*

**Reviewer 1:**
**Overview**
This manuscript describes an analysis of Arctic winter atmospheric properties associated with various characteristic large-scale circulation regimes identified using a clustering tool called self-organizing maps. The authors use reanalysis output, station data, and radiosonde profiles to investigate differences in vertical temperature and moisture profiles, cloud amount and liquid water content, inversion strength and height, and surface energy fluxes associated with varying circulation and advection regimes based on surface pressure. The manuscript is well written, figures are clear, and the analysis is thorough, but the results and conclusions mostly confirm what is already known and do not provide substantial new understanding. In general, they find that conditions in which a warm, moist flow from open-ocean areas penetrates the Arctic are associated with increased cloudiness, increased liquid water content, increased downwelling longwave radiation, and weaker inversion strengths. Circulation regimes that cause flows from ice-covered or land areas tend to be drier with lower cloud amounts, stronger inversions, and smaller downwelling longwave fluxes. The only surprising finding to me was the higher altitude of maximum temperature and specific humidity values in conditions of stronger inversions; I would have expected the opposite. Nevertheless, the study provides a very instructive summary of a complex environment that would likely be useful for educational purposes and for ongoing studies to understand processes associated with the rapid changes that are underway in the Arctic. For these applications, I would support publication. The study would have been much more interesting and enlightening if the authors had explored differences in atmosphere/regime linkages during the "cold Arctic" period (say, 1979-1996) versus the recent decades presented in this manuscript (2009-2018).

*We thank for this summary. As noted by the reviewer, this study provides a comprehensive analysis of linkages between atmospheric circulation and vertical thermodynamic profiles. These linkages have previously not been systematically and comprehensively addressed. We decided to focus on the recent pan-Arctic climate to provide this overview, to focus on the interactions and processes and to set the stage for more targeted change studies that would be a topic for future paper.*

**Specific comments**
1. Section 4.2: It's unclear whether Fig. 2 is based on all days in the data set or a subset.
*Figure 2 is based on all days in the data set for the winter months (DJF) and shows median values. Anomalies related to different circulation types are then calculated with respect to these median values. The text has been slighly modified to clarify that Fig. 2 shows winter medians.*

2. In Figure S2, why is the high fraction of liquid water collocated with high pressure in types 1, 3, and 10 but much less so in type 12?
*In all of these four types, it is the warm advection, typically from south, which increases the fraction of liquid water. The role of a high pressure is to steer this advection. Hence, depending on the orientation of a high pressure and direction of related advection, a high pressure can either be linked to a higher or lower fraction of liquid water. The role of advection for fraction of liquid water is discussed in the manuscript.*

3. Line 224: change shortly to briefly
*Changed.*

4. 203-204: In my experience, using anomaly fields to create the SOM eliminate the problem of all types exhibiting some of the same features. Anomaly fields accentuate differences among types and often assist in interpreting other fields mapped to the SOM. It would be interesting to see if results changed if anomalies had been used to create the SOM, and how different the types would look compared to Fig. S1.
*To test this we ran the SOM analysis with the anomalies and could verify that the results are the same as in Fig. S1.*

5. 224: change shortly to briefly
*Changed.*

6. 268-273: Is ERA5 able to capture realistic differences in cloud phase? This seems an important question given the sensitivity of surface fluxes to this variable.
*This is a very relevant question. The cloud scheme used in the version of IFS, on which ERA5 is based, was dramatically upgraded compared to that ERA-Interim and now has five prognostic variables; liquid and frozen, cloud and precipitation water (Forbes et al. 2011) and cloud fraction (Tiedtke 1993), such that the evolution of cloud liquid water and ice are carried in separate but coupled forecast equations; this new scheme was further improved for Arctic mixed-phase clouds in Forbes and Ahlgrimm (2014). The scheme was tuned so that at temperatures below -38°C, all clouds consist of ice only, -38°C being the threshold for homogenous freezing, but allowing for a dynamic adjustment of liquid and ice depending on the physics and dynamics of each situation.*

*Unfortunately, due to lack of observational data, it is not possible to verify how realistically ERA5 represents cloud phase over the pan-Arctic domain. The scheme was developed using the so-called supersites across the globe (e.g. the ARM and NOAA sites in Utqiagvik) and the so-called A-Train of active satellite sensors. However, reliable winter time data north of 80°N (the northern limit for the active satellite sensors), especially for low clouds (below 0.5 km) besides the super sites simply does not exist. Hence, when evaluating subtleties in cloud water over larger areas, ERA5 is what there is. We must therefore assume that the uncertainty related to cloud phase is high, and this has been mentioned in the revised manuscript.*

*References:*
*Tiedtke, M. (1993) Representation of clouds in large-scale models. Monthly Weather Review, 121(11), 3040– 3061.*

*Forbes, R. M., A. M. Tompkins and A. Untch, 2011: A new prognostic bulk microphysics scheme for the IFS, ECMWF Technical Memoranda 649, Available at: https://www.ecmwf.int/en/elibrary/9441-new-prognostic-bulk-microphysics-scheme-ifs*

*Forbes, R.M. and Ahlgrimm, M. (2014) On the representation of high-latitude boundary-layer mixed-phase cloud in the ECMWF global model. Monthly Weather Review, 142(9), 3425– 3445.*

7. Section 4.3: I'm surprised that more attention has not been given to precipitable water (total column water vapor), as it has a large impact on downwelling longwave fluxes in winter, especially when clouds are absent.

*The variations in surface net radiation variations between different circulation types are tightly connected to distributions of total cloud water in ERA5. While precipitable water has an impact on downwelling radiation, this impact is most significant in clear-sky conditions. However, clear-sky conditions are quite rare in ERA5 in winter; at many locations occurring less than 10% of the time (Fig. 2a and c). Hence, because of the dominance of cloudy cases in ERA4, and this may well be an example of the uncertainty discussed above, impacts of precipitable water are not easily distiguihable in our results. Instead of precipitable water, representing the total amount of water vapour in an air column, we address vertical profiles of specific humidity (in Section 4.4), which provide much more detailed picture of moisture conditions in the atmosphere.*

8. 296: This statement may not hold as the sea ice becomes much thinner and broken.
*It is true that this holds for rather high sea ice concentrations, not for example for conditions at the marginal ice zone. Hence, we have added to the manuscript that this statement refers to high-concentration sea ice. Because this article addresses recent climate, we will not refer to future changes in sea ice here.*

9. Figures 4 e-p: It's very difficult to see important differences among types. I suggest instead plotting differences from median values to make differences more conspicuous. The color scale used for m-p is also not optimal for displaying differences.
*It is important for interpretation of the results to clearly display the directions (upward/downward) of fluxes; directions are not clear from anomalies. Therefore, after careful consideration, we have decided to only show median values instead of anomalies for fluxes in Fig. 4. This decision is argued in the beginning of Section 4.3.*

*We agree that the color scale for 4 m-p was not optimal, and we have therefore changed the color scale in this revised version of the manuscript to display the differences better.*

10. 322: I suggest adding "and resulting inversion strength" after "by the surface".
*We have now added "and resulting inversion formation" here.*

11.323-325: How accurate are the elevations of T and q maxima? Is the difference between the heights statistically significant?

*In this study, we utilize the full vertical resolution model level data of ERA5. Elevations of T and q maxima are directly taken from this model output, without interpolation or averaging. In that sense, these maxima are exact and accurate values as provided by ERA5. Comparison to radiosoundings provides an estimate of realistic these ERA5 elevations are. As it can be seen in Fig. 5, radiosoundings and ERA5 mostly agree. The height difference between T and q maxima, to which we refer to in the manuscript text, is also visible in the results based on radiosoundings. Hence, we consider these estimates reliable. However, for the Arctic Ocean observational data are not available.*

*The statistical significance was tested by applying Kolmogorov-Smirnov test. The test determines if two samples are from the same parent distribution. The difference between heights of T and q maxima were*

*statistically significant at the 95% confidence level practically everywhere, except over open ocean; this information has been added to the text.*

12. 325-327: I suggest providing the standard deviation to give a sense of variability in these values.
*Standard deviations are now given here.*

13. 332-335: It's surprising that ERA5 and raobs differ given that the raobs are assimilated into the reanalysis. Perhaps type 1 collects a relatively wide variety of circulation patterns, which is typical for corner nodes of the SOM.

*Assimilation of radiosoundings to ERA5 is affected by synoptic scale circulation (Hersbach et al. 2020), suggesting that for some circulation types ERA5 and radiosoundings probably agree more than for others. Circulation type 1 is dominated by the Icelandic low and a high pressure over North America, whereas mean seal level pressure (MSLP) features elsewhere have less weight in this circulation type, and some of those are probably averaged out. Generally, anomalies of temperature and humidity profiles in circulation type 1 are weak. We assume that due to these weak anomalies the results for ERA5 and radiosoundings partly disagree. This part in the manuscript has been revised to clarify that the disagreement is probably due to weak anomalies as follows: "Elsewhere in the Arctic anomalies are weaker, probably because this circulation type is dominated by the Icelandic low and a high pressure over North America and features elsewhere are partly averaged out."*

*References:*

*Hersbach, H., Bell, B., Berrisford, P., Hirahara, S., Horányi, A., Muñoz-Sabater, J., Nicolas, J., Peubey, C., Radu, R., Schepers, D., Simmons, A., Soci, C., Abdalla, S., Abellan, X., Balsamo, G., Bechtold, P., Biavati, G., Bidlot, J., Bonavita, M., De Chiara, G., Dahlgren, P., Dee, D., Diamantakis, M., Dragani, R., Flemming, J., Forbes, R., Fuentes, M., Geer, A., Haimberger, L., Healy, S., Hogan, R. J., Hólm, E., Janisková, M., Keeley, S., Laloyaux, P., Lopez, P., Lupu, C., Radnoti, G., de Rosnay, P., Rozum, I., Vamborg, F., Villaume, S., and Thépaut, J.-N.: The ERA5 global reanalysis, Quart. J. Roy. Meteorol. Soc., 146, 1999-2049, doi:https://doi.org/10.1002/qj.3803, 2020.*

14. 360-361: Can you offer an explanation for this finding?
*A probable explanation is the uninterrupted surface cooling and associated air mass ageing. We have now added this explanation to the manuscript text: "The vertical maxima of temperature are found several hundreds of meters higher than the median over the Eurasian Arctic and the Laptev, East Siberian and Chukchi Seas (Fig. 6o), presumably due to the uninterrupted surface cooling allowing for vertical growth of inversion layers."*

15. 370: "does not have any unambiguous impacts" is a convoluted statement – can you reword to clarify?
*Changed to "has ambiguous impacts".*

16. 412: How are temperature gradients defined/measured?
*The following definition is now added to Methods -section (Section 3.2): "Temperature and specific humidity gradients at these stations were derived from thermodynamic profiles by linearly interpolating the profiles to a dense equidistant vertical grid and calculating the difference between adjacent levels."*

17. 419: change largely to greatly

*Changed.*

18. 435: Please explain why a different set of SOM types from the rest of the analysis are used for this location.
*The 12 circulation types represent distinctive MSLP patterns at the pan-Arctic scale, but several types results in similar flow conditions when examining a certain location. Therefore, and to keep the amount of subplots down, we only show four main circulation types for each of the stations, even if analysed the conditions in all 12 of them. For each station we select those which have distinct large-scale flows at that particular location and has associated clear signals in temperature and humidity profiles. As the location is different this means that we had to select a different set of flow patterns.*

*This was already briefly mentioned in Section 4.5.1 for Utqiaġvik, but we have now added this explanation also for Sodankylä. We did not include all the "corner" circulation types (1,3,10 and 12), because they do not optimally represent the variety of large-scale flows at Utqiaġvik and Sodankylä. In the revised version of the manuscript, we write in the Section 4.5.2 that "For Sodankylä in Northern Finland (see location in Fig. 9 a–d), we explore circulation types 12, 1, 4, 5 (Fig. 9), which are distinct and well represent variety of large-scale flow patterns at this location. The different choice of flow patterns compared to for Utqiaġvik is due to the difference in location relative to the flow." We have now added similar text to description for Utqiaġvik in Section 4.5.1.*

19. 440-441: Could this difference just be due to the different types selected for analysis?
*These differences are not due to the different selection of circulation types shown in the manuscript. The differences were evident also when all the 12 circulation types at these stations were compared. In any case, a certain circulation type is associated with very different large-scale weather conditions at Utqiaġvik and Sodankylä, respectively, because of their geographical locations. We have now added to the text "this is evident also when all the 12 circulation types are compared (not shown)"*

20. 470-471: Perhaps the duration of the high pressure is also a factor.
*We agree. The sentence has been modified to include persistence as a factor.*

**Reviewer 2:**

The current climate change in the Arctic is characterised by an unprecedented near-surface warming exceeding the warming in the mid-latitudes by about 2 K and related sea-ice retreat (Arctic Amplification). Arctic Amplification is both a consequence and a driver of local and remote feedback processes specific to the Arctic, but currently there is no consensus on how much the individual feedbacks contribute to AA nor consensus on how AA is linked with the weather and climate in the lower latitudes. Recent studies (e.g. He et al., 2020) suggest that the mid-latitude response to AA depend on the depth of Arctic warming.

Given this background, the submitted manuscript is very valuable since it studies the wintertime thermodynamical structure of the Arctic atmosphere and the impacts of local processes (cloud processes and associated radiative cooling) and large-scale processes (in terms of specific atmospheric circulation patterns) on temperature and specific humidity profiles in the circumpolar Arctic. In addition the manuscript includes a valuable evaluation of the ERA5 reanalysis over the Arctic with respect to cloud cover and temperature and humidity profile metrics at about 30 selected sounding stations.

Therefore, the study should be published in 'WCD'. Nonetheless, at this stage, the submitted manuscript needs careful and major revision.

*We thank for this summary and appreciate that the reviewer considers this manuscript very valuable.*

**Major comments:**

(1) The SOM clustering algorithm has been applied for the classification of the preferred atmospheric circulation pattern. During the last years, this method has been well established for that purpose, but as as all clustering algorithms, it requires the prescription of different parameters.

The authors have to better explain, which criteria they have used to determine the quality of the clustering, and how sensitive are the results of the SOM clustering to the choice of the region (e.g. in comparison to a region north of 50 degrees N) and the choice of the SOM parameters. In particular I wonder why a 3x4 SOM array was choosen, and why only 4 circulation patterns were discussed at a time in sections 4.2 to 4.5. I understood that the 4 patterns analysed in section 4.2 to 4.4 have been choosen to cover the whole range of sensitivities to circulation changes, assuming that this is best covered by the corner patterns of the 3x4 SOM array. I think this assumes implicitely, that the corners of the 3x4 SOM array map to the corners of the corresponding Sammon map (which visualizes the closeness of nodes in a 2D space). I wonder whether the authors have proven that.

*In this study, the main focus is to detect clear signals in temperature and humidity profiles linked to circulation. For this purpose, we needed to cluster atmospheric circulation to extract well-defined dynamical conditions. For an optimal clustering, we needed to have large enough (≥ 2x3) SOM array, in which there is not too much variability between individual cases belonging to a circulation type (to avoid averaging out some of the features in circulation fields). In the initial phase of the study, we made sensitivity tests with the array size (3x4, 4x5, 5x6), and based on these tests, 3x4 was found to be the most appropriate array size for this study. The 3x4 generalizes the circulation, but importantly, it has still enough details to allow identification of clear anomalies in temperature and humidity profiles. From our experience, the analysis is not very sensitive to the choice of region. For sensitivity tests related to*

*domain size as a part of other ongoing studies, we have found out that shifting the southern boundary towards south (e.g. to 45°N) has a very small effect compared using the present SOM domain. The main difference with a more southern boundary is that the Aleutian low gets more weight in the circulation types but this is not very essential for our study.*

*We want to emphasize that we have analyzed results for all 12 circulation types. However, when preparing the manuscript, we wanted to condense the results to a clear story, without exhausting a reader by showing and presenting results for all the 12 circulation types. There is also a difference in examining pan-Arctic flow patterns and comparing details at specific locations. For a given location, several flow patterns results in quite similar regional or local flow; hence the number of circulation types can be reduced for clarity.*

*For this reason, we picked 4 circulation types to demonstrate the dependencies between circulation, clouds, fluxes and thermodynamic profiles. These 4 circulation types are first and foremost examples of these dependencies and they demonstrate the general relationships with the dynamics, which are also commonly found in the other circulation types. We chose to focus on the types located in the corners of the SOM array because they represent clearly distinct circulation regimes which are also among the most common ones.*

*We agree that a Sammon map would provide quantitative information about the relationships between circulation types in our SOM array. We have not made a Sammon map of our SOM array, because for this study relationships between circulation types themselves and their quantitative closeness are not relevant. Instead, it is relevant to demonstrate spatial distributions of clouds, fluxes and thermodynamic profiles in well-defined dynamical conditions. The general conclusions of this study are not dependent on the choice of example circulation types shown.*

*In this revised manuscript, we have added to Section 3.1. that "Sensitivity tests indicated that relationships between large scale circulation and thermodynamic profiles are insensitive to the size of the SOM array and changes in the size of the study domain." At the end of Section 4.1 we have added a clarification why we show these four circulation types: "These four circulation types demonstrate the general dependencies between circulation, clouds, fluxes and thermodynamic profiles, which are also commonly found in the other circulation types."*

In 4.5 (the analysis of two specific Arctic stations) the 4 circulation pattern which have been chosen to cover a wide range of circulation sensitivities have been selected with respect to the specific position of the station (as far as I understood). I recommend that the authors explain this in more detail, and discuss if this selection of the 4 circulation patterns out of the 3x4 SOM array could be done in a more objective manner.

*The 12 circulation types represent distinctive MSLP patterns in the scale of the whole Arctic, but not at individual stations. In one hand, several of the types pose rather similar flow conditions for a certain location. On the other hand, because of the different geographical locations of Utqiaġvik and Sodankylä, particular circulation types can be associated with very different large-scale weather conditions at these locations. Therefore, for both of the stations we only show four circulation types, which have distinct large-scale flow at the station and associated clear signals in temperature and humidity profiles. This was already briefly mentioned in Section 4.5.1 for Utqiaġvik, but we have now added this explanation also for Sodankylä. The four circulation types shown for each station were selected to represent variety of large-scale flows at the station. The results presented through these four examples highly reflect the dynamical relationships and features found in all 12 circulation types. It is very difficult to find an objective way to*

*choose which circulation types to present in the manuscript. Importantly, even with an objective way to select the four circulation types for the stations, the main results and conclusions would be the same. We note that these issues were also raised by Reviewer 1. See also our response to her/him.*

*In the revised version of the manuscript, we write in the Section 4.5.2 that "For Sodankylä in Northern Finland (see location in Fig. 9 a–d), we explore circulation types 12, 1, 4, 5 (Fig. 9), which are distinct and well represent variety of large-scale flow patterns at this location." Accorrdingly, we have now written to description for Utqiaġvik in Section 4.5.1: "…These are distinct and represent the effects of variety of large-scale flows at this location well."*

(2) I appreciate the efforts of the authors to divide the Arctic into 5 regions with specific characteristics, and to summarize the results in this way. To make the linkages to the circulation patterns even clearer and easier to follow for the reader, I recommend to describe the linkages more explicitly throughout the whole section, not only for region (4) at L507.
*Following this suggestion, we have now added direct linkages to specific circulation patterns for all the 5 regions presented in Discussion section.*

Furthermore I wonder, if the temporal variations of temperature inversions over Greenland (see L 521) would show a stronger association with specific circulation patterns, if other circulation patterns than the corner patterns would have been included in the analysis. I assume a stronger association of temperature inversions over Greenland with the circulation pattern typ 4 which displays a strong Icelandic low, probably as part of a NAO+ pattern.
*We emphasize that we have analyzed bulk temperature inversions and other profile metrics in all 12 circulation types (see circulation types in Fig. 1, and bulk temperature inversion anomalies in Fig. R1 below), even if only the results for 4 circulation types are shown in the manuscript. As the reviewer assumes, in circulation type 4 (as well as in types 5, and 6) there are positive anomalies of bulk inversion strength associated with the strong Icelandic low. However, type 9 with a relatively strong Icelandic low with the approximately same location is associated with mostly negative bulk inversion anomalies over Greenland. Hence, we agree that there is some association with the circulation patterns also over Greenland, but they are less clear and consistent than for many other Arctic regions.*

*We have revised the text in the manuscript. The new version is: "In our results, we also see this spatial distribution within Greenland, and note that the strongest temperature inversions in the northern Greenland are often associated with a strong Icelandic low south of Greenland (types 4, 5 and 6 in Fig 1)."*

[Figure]

*Figure R1: Anomalies of vertical maximum T–surface T (bulk inversion strength) in the 12 SOM circulation in December–February 2009–2018. The colours inside the circles show values based on radiosonde observations, whereas the fields are based on ERA5. The dotted shadings (ERA5) and black circles (radiosonde observations) show the that are different from zero at the 99% significance level (only for types 1,3, 10 and 12).*

(3) It is clearly stated in the manuscript that the effects of large scale circulation are manifested as heat and moisture advection. So far, anomalous heat and moisture transport related to the different circulation changes are discussed only implicitly on the basis of the mean sea-level pressure patterns. I suggest to include an additional metric which quantifies the anomalous advection more directly (anomalous horizontal wind fields or, if feasible, even anomalous temperature and moisture fluxes).

*This is a very relevant comment. We have followed this suggestion and included horizontal heat and moisture transports in the manuscript; a new figure is in the Supplemental material and we refer to that figure when we write about transport. Instead of anomalies, we show the median values which directly indicate the direction of transport.*

(4) I recommend to improve the figures for the gradient profiles. For me it was very difficult to see the differences between all the profiles (and therefore to follow the description and arguments in section 4.5), and I wonder, whether they could be moved to the supplement.

*Because vertical gradients cannot be directly seen from median and percentile temperature and specific humidity profiles in Figs. 8 and 9, we had also included the gradient profiles. They better visualize shapes of individual profiles. We, however, admit that figures of the gradient profiles are not very clear. Because it was challenging to make them visually clearer, we decided to move the to the Supplements, as suggested by the reviewer.*

**Minor comments**

(1) The authors should give a brief explanation why they analysed the period 2009-2018 only.
*In this study, we focus on processes and interactions in the present climate and the recent ten-year period (2009-2018) is a sufficiently long "sample" of the current climate. Long-term/decadal changes of these interactions are out of the scope of this study. We have added an explanation to the last paragraph of Introduction: "The focus is on processes and their interactions in the present climate."*

(2) L89-91: Another approach for comparing station data with reanalysis is to use the average of reanalysis data from the 4 closest grid points surrounding the station instead of using the closest grid point only. Why did the authors used the latter approach? And just for my curiosity: in the case that the authors compared the two approaches, what was the outcome of that excercise?
*This formulation was a mistake and has now been omitted in the revised text. This sentence was referring to comparisons specifically made for Utqiaġvik and Sodankylä that were finally excluded from the submitted version of the manuscript. We are grateful that the reviewer helped us to spot this mistake.*

(3) L134-145: Please mention the region used for the SOM clustering explicitely here.
*Thanks for pointing out that this relevant information was missing. We have now added here that the SOM analysis was made for the region north of 50°N.*

(4) L213-214: "...in which the Icelandic low has a relatively western location." This statement is not valid for type 10, please correct.
*Corrected to "…while the most persistent types (i.e., with the longest uninterrupted duration) are 1, 4, 9 and 10 (Fig. 1), in which the Icelandic low has a relatively western location (types 1, 4 and 9) or there is a strong high pressure over the Arctic Ocean (type 10)."*

(5) L249: Fig. 3e and i --> please correct to Fig. 3e and m!
*Corrected.*

(6) L363-364: Please be more precise on the region (westward from 60N over land or ocean??), otherwise it is difficult to follow the discussion.

*We have added "over land" here to specify the region.*

---

## Author Response (AR2)

**Dear Editor and Reviewer,**

**We highly appreciate the further suggestion to improve our manuscript. Our response is written in italics below the comments.**

**Comments by the editor:**

I would like to thank the authors for their revisions to the paper. The reviewer has suggested that some more evidence could be given to justify the choice of SOM array size. I agree that this would be useful information which would improve the paper, and that a further minor revision should be sufficient to include this. I look forward to seeing a revised version of this paper in due course.

**Comments by the reviewer:**

I very much appreciated the efforts made by the authors to improve the manuscript. In my view, I have only one minor comment which is related to the answer to my previous major comment (1).

I appreciate that the authors performed sensitivity tests with different domain sizes and different sizes of the SOM array. The authors claimed that they "made sensitivity tests with the array size (3x4, 4x5, 5x6), and based on these tests, 3x4 was found to be the most appropriate array size for this study." and that "the analysis is not very sensitive to the choice of region".

In my view, these statements are rather subjective. I recommend that the authors provide objective measures/metrics for the different SOM analyses which give more evidence for these statements.

Although the SOM technique provides an objective method for clustering, the choice of the SOM array size is always somewhat subjective (Alexander et al. 2010). Kohonen et al. (2014) (the first author being the developer of the SOM method) stated that the SOM array size must be determined by the trial-and-error method, after seeing the quality of the first guess. The subjectiveness of selection of SOM array size is widely recognized and accepted by the scientific community, and we have limited possibilities to overcome this feature of the method. The size of an optimal SOM array depends on the intended application and the size of the data space spanned by the input data (Cassano et al. 2006).

Impacts of SOM array size have been tested in various studies (e.g., Cassano et al. 2006, Cassano et al. 2007, Higgins and Cassano 2009, Liu et al. 2006, Nigro and Cassano 2014, Skific et al. 2009). These studies have provided qualitative assessments of array size, mostly based on visual inspection. They all agree that a larger number of nodes provides greater detail (of circulation types), while a smaller number of types represents the archetypical types with few details.

The reviewer recommended us to provide objective measures for the impacts of the array size and region size. According to our knowledge, purely objective measures for those do not exist. Cassano et al. (2015) estimated the effects of SOM array size by calculating Root-Mean-Square-Difference (RMSD) and twistedness index (TI) in different sized SOM arrays; however, also their approach included subjective assessement to select which characteristics of pressure patterns should be represented in the final SOM array and, in particular, what are the accepted/desired values of RMSD and TI as there are no subjective tresholds for those. Alexander et al. 2010 calculated sum of Root-Mean-Square Euclidean distances between the SOMs and the target dataset to find the smallest errors. Their final choice of SOM array was

a compromise between minimizing errors and providing sufficiently different synoptic patterns to be useful for a climate change study, thus also based on a subjective assessment.

To compare the impacts of SOM array sizes, we show here the mean sea level pressure fields in 2x3, 3x4 and 4x5 sized SOMs in Figures R1–R3.

Figure R1. Mean sea level pressure fields in 2x3 SOM